## Original Research Article

agent-based modeling; cortex endodermal initial divisions; hybrid multi-scale modelling; ordinary differential equations; root stem cell niche; WOX5.

**Author for correspondence:**
R. Sozzani,
E-mail: ross_sozzani@ncsu.edu

# A hybrid model connecting regulatory interactions with stem cell divisions in the root

Lisa Van den Broeck[1], Ryan J. Spurney[2], Adam P. Fisher[1], Michael Schwartz[1], Natalie M. Clark[3], Thomas T. Nguyen[1], Imani Madison[1], Mariah Gobble[1], Terri Long[1] and Rosangela Sozzani[1]

[1]Plant and Microbial Biology Department, North Carolina State University, Raleigh, North Carolina, USA; [2]Electrical and Computer Engineering Department, North Carolina State University, Raleigh, North Carolina, USA; [3]Department of Plant Pathology and Microbiology, Iowa State University, Ames, Iowa, Iowa 50010, USA

## Abstract

Stem cells give rise to the entirety of cells within an organ. Maintaining stem cell identity and coordinately regulating stem cell divisions is crucial for proper development. In plants, mobile proteins, such as WUSCHEL-RELATED HOMEOBOX 5 (WOX5) and SHORTROOT (SHR), regulate divisions in the root stem cell niche. However, how these proteins coordinately function to establish systemic behaviour is not well understood. We propose a non-cell autonomous role for WOX5 in the cortex endodermis initial (CEI) and identify a regulator, ANGUSTIFOLIA (AN3)/GRF-INTERACTING FACTOR 1, that coordinates CEI divisions. Here, we show with a multi-scale hybrid model integrating ordinary differential equations (ODEs) and agent-based modeling that quiescent center (QC) and CEI divisions have different dynamics. Specifically, by combining continuous models to describe regulatory networks and agent-based rules, we model systemic behaviour, which led us to predict cell-type-specific expression dynamics of SHR, SCARECROW, WOX5, AN3 and CYCLIND6;1, and experimentally validate CEI cell divisions. Conclusively, our results show an interdependency between CEI and QC divisions.

## 1. Introduction

Stem cells divide to regenerate themselves and to generate all of the cell- and tissue-types in a multicellular organism, such as plants. The continued ability to sustain stem cells within their micro-environment, the stem cell niche (SCN), is an important developmental characteristic that ensures proper tissue growth. The *Arabidopsis thaliana* root SCN contains four stem cell populations, the columella stem cells (CSCs), the cortex endodermis initial (CEI) cells, the vascular initial cells and the epidermal/lateral root cap initials, which form the entire root as a result of consecutive cell divisions (Dinneny & Benfey, 2008; Fisher & Sozzani, 2016). The different populations of stem cells are maintained by the quiescent center (QC) through the generation of short-range signals that repress cell differentiation (Clark, Fisher, et al., 2020; Pi et al., 2015; van den Berg et al., 1997). A known QC-derived signal is the homeobox transcription factor (TF) WUSCHEL-RELATED HOMEOBOX 5 (WOX5), which is specifically expressed in the QC and represses the differentiation of the CSCs (Petricka et al., 2012; Sarkar et al., 2007). Specifically, non-cell-autonomous WOX5 maintenance of CSCs takes place through the repression of the differentiation factor *CYCLING DOF FACTOR 4* (Pi et al., 2015). *wox5-1* mutants have increased QC divisions in roots and a decreased number of columella cell layers (Forzani et al., 2014). In the QC cells, WOX5 controls divisions by restricting *CYCD3;3* expression (Forzani et al., 2014). Although the regulatory modules within the CSCs and QC are well characterized (Forzani et al., 2014; Stahl et al., 2013), the molecular mechanisms by which WOX5 promotes stem cell fate of CEIs remain unknown.

Several proteins have been shown to positively regulate *WOX5*, such as ANGUSTIFOLIA (AN3)/GRF-INTERACTING FACTOR 1 (GIF1). *AN3* is expressed in the root meristem with a high peak in expression in the SCN and QC and plays a role in maintaining QC identity (Ercoli et al., 2018). However, whether AN3 function is dependent on WOX5 and whether AN3 has a regulatory role outside the QC in the SCN is not understood. Additionally, AN3 was

shown to regulate the expression of *SCARECROW* (*SCR*) (Ercoli et al., 2018), which along with SHORTROOT (SHR) regulates the expression of the D-type Cyclin *CYCLIND6;1* (*CYCD6;1*) to control the CEI divisions to generate the cortical and endodermal tissue layers (Cruz-Ramírez et al., 2012; Gallagher & Benfey, 2009; Long et al., 2015; Nakajima et al., 2001; Sozzani et al., 2010). Specifically, SHR moves from the vasculature to the CEI, where it forms a complex with SCR to transcriptionally regulate CYCD6;1.

The regulatory interactions between the different cell types of the root SCN are complex and non-intuitive, so computational tools are essential to understanding systemic behaviour. Developmental processes such as auxin flow within the root and lateral shoot branching have been mathematically modelled to better understand and predict system-level behaviour (Canher et al., 2020; Prusinkiewicz et al., 2009). Some models implement different scales of the system to simulate, understand and predict system-level behaviour as a whole. For example, a mathematical model that simulates and predicts the induction of shoot branching during plant development included on a molecular scale auxin flux across metamers (i.e., smaller segments of the stem) and on an organ scale the formation of metamers of the stem and lateral branches (Prusinkiewicz et al., 2009). Modeling systems and allowing exchange of information across different scales can also be achieved by combining agent-based models (ABM) with continuous models, such as ordinary differential equations (ODEs) or partial differential equations (Cilfone et al., 2015). ABMs consist of autonomous 'agents' that dynamically interact and show responsive behaviour through a set of simple rules. ABMs have, for example, been used to simulate plant-herbivore interactions (Radny & Meyer, 2018). However, within the molecular plant biology field, these models are not widely used, despite their capacity to capture system-level behaviour. On the other hand, continuous models such as ODEs have been applied to infer gene regulatory networks (Krouk et al., 2010; Yao et al., 2011) and predict dynamic gene expression patterns (Clark, Fisher, et al., 2020). These models are computationally intensive and lack the capability to capture system-level behaviour but can model complex dynamic responses over time. Hybrid models are created when, for example, continuous models are used within a discrete ABM to describe a part of the system. These hybrid models are usually multi-scale models, given that the continuous models often describe a dynamical response on a different spatiotemporal scale than the ABM (Cilfone et al., 2015).

In this study, we combine cell-type-specific gene expression data and experimental data with network inference and parametric models to better understand how WOX5, AN3, SCR, and SHR coordinately regulate CEI stem cell divisions. We transcriptionally profiled CEI cells in wild-type and *wox5-1* roots, as well as QC cells and non-stem cells. We found that AN3 was among the most CEI-enriched genes. Additionally, the loss-of-function of *wox5* or *an3* resulted in an extended expression pattern of the CEI stem cell marker *CYCD6;1* into the cortex and endodermal cells. We built an ODE and agent-based hybrid model linking cell behaviour, specifically cell division, to gene expression dynamics represented by ODEs of WOX5, AN3, SCR, SHR, and CYCD6;1. Our hybrid model allowed for the exchange of information between a cellular scale (i.e., division of stem cells) and a molecular scale (i.e., regulatory interactions at single cell level). In the hybrid model, the mobile proteins, WOX5 and SHR, regulated the expression of downstream proteins non-cell autonomously in specific cell-types. The communication between cell types and dynamic expression patterns modelled experimentally validated temporal stem cell divisions.

## 2. Results

### 2.1. WUSCHEL-RELATED HOMEOBOX 5 regulates cortex endodermis initial-specific genes

The functional role of WOX5 in the QC and CSC has been extensively reported while its role in stem cell populations remains largely unknown. *WOX5* is specifically expressed in the QC cells, however, the protein moves to the CSCs and the vasculature initials and has been shown to have a non-cell autonomous role in these cells (Clark et al., 2019; Pi et al., 2015). To determine whether WOX5 is also able to move from the QC cells to the QC-neighbouring CEI cells and regulate downstream targets, we used scanning fluorescence correlation spectroscopy (scanning FCS). Five-day-old *wox5*xpWOX5:WOX5-GFP plants were analysed with scanning FCS to evaluate the directional movement of WOX5 proteins between these two cell types. Line scans were taken over time from a region spanning the CEI and adjacent QC (Figure 1a). This analysis resulted in a quantitative assessment of movement and allowed us to calculate the movement index (MI). We found that WOX5 moved bidirectionally between the QC and the CEI (MI = 0.90 ± 0.04 from QC to CEI, MI = 0.83 ± 0.05 from CEI to QC, *n* = 20) (Supplemental Table 1). As a comparison, within the SCN, free GFP and immobile 3xGFP have a moving index of ~0.7 and ~0.25, respectively (Clark et al., 2016).

To explore the potential functional role of WOX5 in CEI, we examined the expression pattern of the CEI-marker pCYCD6;1:GUS-GFP in *wox5*. The marker showed an expression pattern that extended into the cortex and endodermal cells (Figure 1b,c). This expanded expression of *CYCD6;1* suggests that the 4–5 cells proximal of the CEI, further referred to as CEI-like cells, have gained stem cell-like characteristics and also indicates that WOX5 controls *CYCD6;1* expression to the CEI (Figure 1b,c). We then explored the role of WOX5 in limiting *CYCD6;1* expression and, thus, controlling CEI divisions. To this end, we quantified the number of undivided and divided CEI cells in 4-, 5- and 6-day-old *wox5* and wild-type roots. This quantification showed that *wox5*xpCYCD6;1:GUS-GFP roots had an increase of 23.43% and 25.33% divided CEI cells (*p* = .0495, Wilcoxon test) compared to the wild type (WT) at 4 and 6 days, respectively (Figure 1d). Taken together, these results support a functional non-cell autonomous role for WOX5 in the CEI.

### 2.2. Network inference and node importance analysis to identify functional candidates

To unravel the transcriptional events regulating the extended expression pattern of *CYCD6;1* in the *wox5* mutant background, a transcriptome analysis was performed on FACS-sorted GFP positive cells from pCYCD6;1:GUS-GFP, *wox5*xpCYCD6;1:GUS-GFP and pWOX5:GFP, and the meristematic cells from pWOX5-GFP that do not express the marker (referred to as non-stem cells) (Supplemental Table 2). Compared to the cells not expressing the pWOX5-GFP marker, 163 genes were differentially expressed (FDR < 0.05) in wild-type CEI cells and 213 genes in the CEI and CEI-like cells from the *wox5* mutant. In total, the union of these two analyses identified 330 differentially expressed genes (DEGs) in CEI and CEI-like cells, of which 159 DEGs (48.18%) have previously been shown to be expressed in the SCN and 53 genes were enriched in the CEI (Clark et al., 2019). We hypothesized that the regulatory genes underlying *CYCD6;1* expression should be differentially expressed in the CEI cells (*CYCD6;1* expressing cells) of the wild-type and *wox5* roots and thus focused on the genes

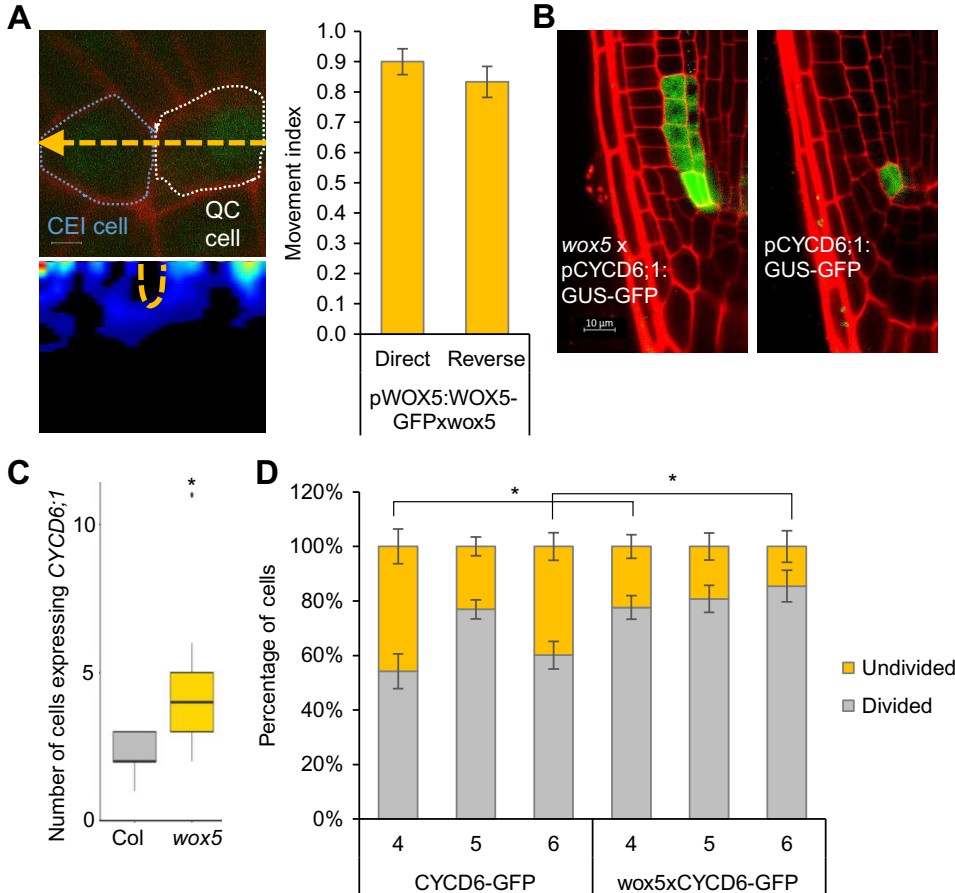

**Fig. 1.** Characterization of WUSCHEL-RELATED HOMEOBOX 5 (WOX5) in the cortex endodermis initial (CEI). (a) (Top left) Confocal image of a region in the *wox5*xpWOX5:WOX5-GFP root that spans the quiescent center (QC) and CEI and is used for pair correlation function (pCF). The location and direction of the line scan (orange dashed line) are marked onto the image. (Bottom left) pCF carpet image of the top image. Orange, dashed region represents an arch in the pCF carpet, which indicates movement across the cell wall. (Right) Movement index of *wox5*xpWOX5:WOX5-GFP between the QC and CEI. (b) Confocal image of *wox5*xpCYCD6;1:GUS-GFP roots. (c) The number of CEI and CEI-like cells expressing pCYCD6;1:GUS-GFP. (d) Percentage of divided and undivided CEI cells in pCYCD6;1:GUS-GFP and *wox5*xpCYCD6;1:GUS-GFP roots. Data are presented as mean ± SEM (standard error of mean). * = $p < .05$ (c, d: Wilcoxon Chi-square test).

overlapping between these two sets of DEGs (Figure 2a). In total, 46 genes overlapped between the CEI and CEI-like cells, which equals an enrichment of 35.8 ($p < 4.431e-59$, Exact hypergeometric probability). To identify key regulatory proteins among these 46 genes, we predicted causal relations between the TFs and downstream genes with high accuracy and constructed a gene regulatory network. We inferred the causal relations by leveraging our transcriptome data with a regression tree algorithm RTP-STAR (Figure 2b) (Huynh-Thu et al., 2010; Spurney et al., 2020; Van den Broeck et al., 2020). The inferred network contained 20 nodes, of which four are TFs (Figure 2b). These four TFs are as follows: WIP DOMAIN PROTEIN 4, which is shown to be important for root initiation, INDOLE-3-ACETIC ACID INDUCIBLE 33, AN3/GIF1, which is a known regulator of cell proliferation, and an unknown TF (AT1G75710). Among the inferred AN3 targets, we confirmed with TChAP data that three targets (*AT1G75710*, *FLA10* and *GBSS1*) were directly bound by AN3 (Vercruyssen et al., 2014). Network inference allowed us to identify potential functionally important genes, however, we still needed to pinpoint the biological important genes within the network.

To identify which genes could cause the largest impact on network stability when perturbed, we performed a node importance analysis. To calculate the impact of each gene, each node received a weight depending on its outdegree (i.e., number of

outgoing edges), then for each node, the sum of the weighted outgoing first neighbours and the sum of the weighted incoming first neighbours were taken. Both sums were in turn weighted, specifically, the sum of the outgoing neighbours was weighted by Average Shortest Path Length (ASPL), and the sum of the incoming neighbours was weighted according to the proportion of end-nodes within the network, which is in this network 20% (see Section 4). We next developed an R-based Shiny application (Node Analyzer) that calculates the weights and impacts of each gene within a network (Shannon et al., 2003) (see Section 4) (Supplemental Figure 1). Node Analyzer allowed us to rank the 20 genes in the network and select key genes. The most impactful gene within our network is *AN3*, a transcriptional co-activator that is involved in cell proliferation during leaf and flower development (Figure 2c).

### 2.3. ANGUSTIFOLIA 3 contributes to the regulation of cortex endodermis initial divisions

It was previously shown that *AN3/GIF1* and its closest homologs, *GIF2* and *GIF3*, were expressed in the root stem cell niche (Ercoli et al., 2018). A triple mutant (*gif1/2/3*) displayed a disorganized QC and increased root length as a result of an increased root meristem size (Ercoli et al., 2018). We confirmed the growth repressing role

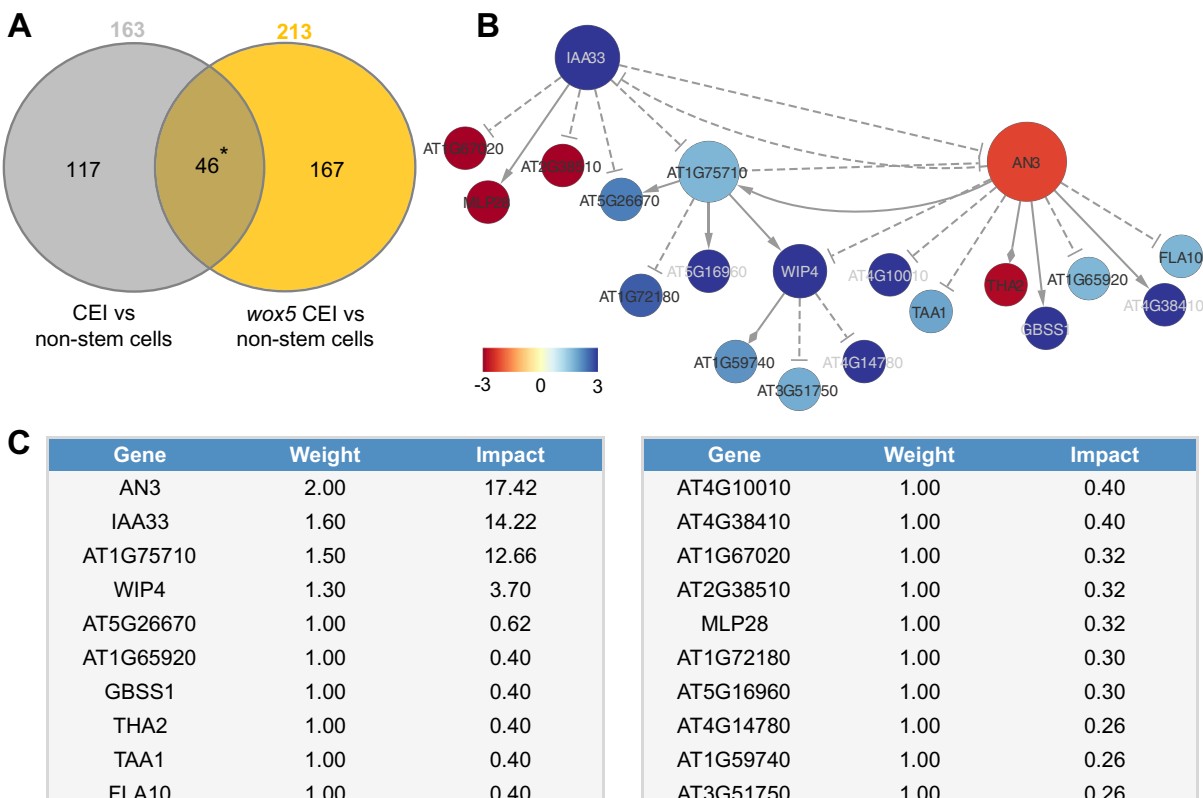

**Fig. 2.** Network analysis of cortex endodermis initial (CEI)-expressed genes. (a) The overlap between genes differentially expressed between CEI cells and non-stem cells in the wild-type and the *wox5* background. $* = p < .001$ (Exact hypergeometric probability). (b) Causal interactions between 46 differentially expressed genes that are enriched in the CEI cells. Solid triangle-arrows, solid diamond-arrows and dashed T-arrows represent activating, undetermined and repressing regulations, respectively. The size of the nodes correlates with the outdegree of that node. The colour of the nodes corresponds to the $\log_2$ fold change in expression in the *wox5* CEI cells compared to the non-stem cells. (c) Tabular output from the Node Analyzer application presenting the weight (calculated based on outdegree) and impact (see Section 4) of each gene.

of AN3 in the roots, as *an3* and 35S:AN3-GFP roots showed an increased and reduced root length compared to the WT, respectively (Supplemental Figure 2a). We observed a disorganized stem cell niche in 56% (25/45 roots) of *an3* mutant roots (Supplemental Figure 2b). Additionally, *an3* mutants contained starch granules in the cells that are normally CSC, suggesting that AN3 plays a role in CSC maintenance (Supplemental Figure 2c). To determine whether AN3 also plays a role in CEI divisions, we quantified the number of undivided and divided CEI cells in 4-, 5- and 6-day-old *an3* and WT roots. Six-day-old *an3* roots had 19.22% fewer undivided CEI cells compared to WT ($p = .103$, Wilcoxon test), suggesting that more CEI divisions occur in the *an3* mutant (Figure 3a). Additionally, when *an3* is crossed with the CEI-marker pCYCD6;1:GUS-GFP, an extended expression pattern is observed (Figure 3b,c). Taken together, these results support a role for AN3 in the regulation of CEI divisions.

### 2.4. A hybrid model to dynamically simulate and predict stem cell divisions

If AN3 and WOX5 are indeed key regulators for CEI divisions, we would expect that their temporal expression influences CEI divisions in a cell-type specific manner. To gain insight into the system-level regulation of CEI stem cell divisions, we modelled the expression of *CYCD6;1* and its direct and indirect upstream regulators: SHR, SCR, WOX5, and AN3 (Figures 1c and 3b) (Sozzani et al., 2010). For this, we developed a hybrid model that

combines agent-based modeling aspects with ODEs. Specifically, we included four different cell types or 'agents' (QC, CEI, vascular initial, and endodermal cell) and constructed ODEs of the genes for each cell type that are able to recapitulate the dynamics of the upstream regulatory interactions at a molecular scale. The cells/agents interact through the movement of SHR and WOX5 and change state (i.e., divide) upon changes in the expression of specific proteins. For example, when CYCD6;1 exceeds a certain abundance, the CEI will divide. Each time a cell divides (an agent changes state), corresponding protein abundances are halved. As such, we were able to exchange information bidirectionally, from molecular to cellular scale and from cellular to molecular scale. To implement this hybrid model, we used SimBiology that models, simulates, and analyzes dynamic systems, allows for rapid model optimization, and provides an intuitive visualization of the model (The MathWorks, 2019).

To analyse the temporal expression dynamics of *CYCD6;1* linked to CEI divisions, and to understand the regulatory role of WOX5 and AN3 in controlling the *CYCD6;1* dynamics, we used ODEs to generate a quantitative model that describes the dynamics of four key transcriptional regulators of *CYCD6;1*, namely WOX5, AN3, SHR and SCR. In our ODE systems, each ODE included a degradation term and a production term that depended on its upstream regulations. The included regulations are depicted in Figure 4 and are as follows: (a) the inhibition of *SHR* by WOX5 in the vasculature (Clark, Fisher, et al., 2020), (b) the activation of *SCR* by the SHR/SCR complex in the endodermis, CEI and QC

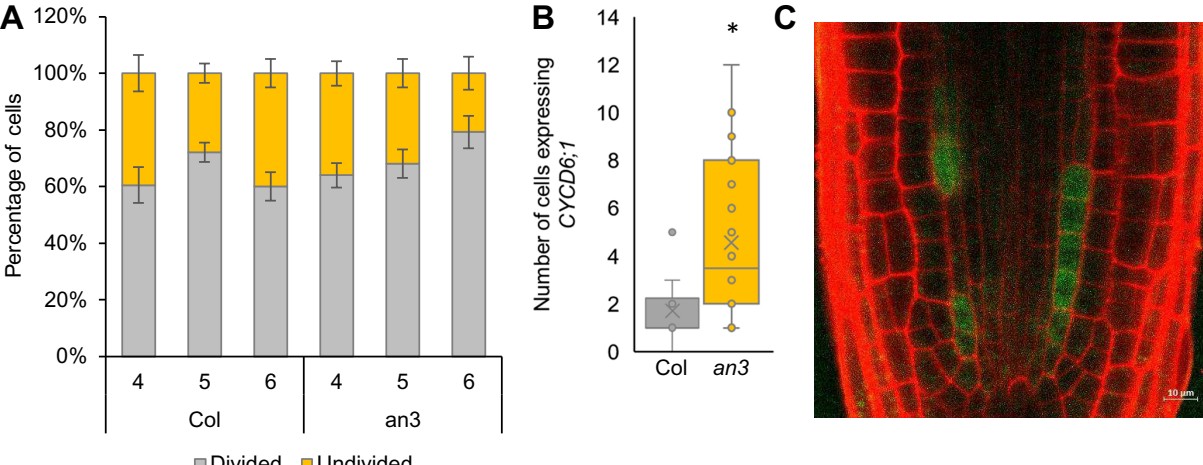

**Fig. 3.** Phenotypic analysis of *angustifolia (3)*. (a) Percentage of divided and undivided cortex endodermis initial (CEI) cells in wild-type (Col) and *an3* roots. (b) The number of endodermal and cortex cells expressing pCYCD6;1:GUS-GFP in wild-type (Col) and *an3*xpCYCD6;1:GUS-GFP roots. (c) Confocal image of an *an3*xpCYCD6;1:GUS-GFP root. Data are presented as mean ± SEM. ∗ = $p < .05$ (a, b: Wilcoxon Chi-square test).

(Heidstra et al., 2004; Helariutta et al., 2000), (c) the activation of *SCR* by AN3 (Ercoli et al., 2018), and (d) the activation of *CYCD6;1* by the SHR/SCR complex in the CEI (Figure 4a) (Sozzani et al., 2010). As the upstream transcriptional regulations of WOX5 and AN3 are unknown, we modelled their expression based on previously published data of *WOX5* and *AN3* expression over time in the SCN (Clark et al., 2019). Additionally, we included ODEs that model the movement of WOX5 from the QC to the vasculature initials (Supplemental Table 3), different diffusion rates of SHR from the vascular initials to the endodermis and QC (Clark, Fisher, et al., 2020), the SHR/SCR complex formation, and the oligomeric states of WOX5 and AN3. The oligomeric states of AN3 and WOX5 were experimentally determined using scanning FCS (Supplemental Figure 3). Specifically, we performed Number and Brightness (N&B) on *an3* or *wox5* roots expressing pAN3:AN3-GFP or pWOX5:WOX5-GFP translational fusion, respectively. We found that both AN3 and WOX5 primarily exist as a monomer (98.67 and 96.01%, respectively) with a very small amount of dimerization (1.33 and 3.99%, respectively) (Supplemental Figure 3). Thus, we fixed the oligomeric state of AN3 and WOX5 as monomers in our ODE model. As SHR and SCR dimers show a similar expression pattern as the monomers (Clark, Fisher, et al., 2020), we simplified the model and reduced the number of parameters by modeling the SHR and SCR monomer and dimer as one variable. Despite this simplification and the experimental estimation of several parameters, the number of parameters in the hybrid model still reaches over 30 as a result of its multi-scale nature spanning both cellular and molecular interactions. To further reduce the number of parameters that needed to be estimated, the most influential parameters were identified with a sensitivity analysis (Sobol', 2001) (Supplemental Table 4, Supplemental Figure 4).

We estimated the values for the sensitive parameters by fitting our model to computed cell-type specific time course data (Supplemental Tables 5–7). Specifically, the expression of the modelled genes in each cell type at 5 days was extracted from cell-type specific datasets (Clark et al., 2019; Li et al., 2016) and overlaid onto a stem cell time course to obtain cell-type specific expression levels every 8 hours from 4 to 6 days (see Section 4) (Supplemental Table 5). After estimating the sensitive parameters, we simulated the hybrid

model to evaluate the expression dynamics within each cell. For example, the hybrid model predicted high expression of *SCR* in the endodermal cells and a lower expression in the CEI and QC. We confirmed the increased SCR expression in the endodermal cells by analysing confocal images of the QC, CEI and endodermal cells of pSCR:SCR-GFP for corrected total cell fluorescence (CTCF) at 5 days 16 hours (Supplemental Figure 5a,b). Model simulations showed that the cell-specific networks ensured robust stability of cellular behaviour, such as cell division regulation (Figure 4b). The agent-based rules for cell division were set based on SHR/SCR complex and WOX5 expression for the QC and CYCD6;1 expression for the CEI (Supplemental Figure 6). Our hybrid model was able to capture a dynamic expression pattern for the SHR/SCR complex, with high expression at 4 days 8 hours and 5 days 16 hours. In contrast, WOX5 showed a low expression at these time points (Supplemental Figure 5c). The first peak of SHR/SCR expression at 4 days 8 hours was previously shown in an ODE model, while the second peak occurred, compared to our model, earlier at 5 days 8 hours (Clark, Fisher, et al., 2020). Model predictions showed that the fine balance between low expression of the SHR/SCR complex and WOX5 simulates a QC cell division at 5 days 5 hours. Indeed, 5- and 6-day-old plants showed an increase in QC divisions compared to 4-day-old plants (Supplemental Figure 5d). Additionally, CEI divisions were predicted to occur at 4 days 8 hours and 5 days 16 hours (Figure 4b). We observed an increased percentage of divided CEIs in 5-day-old roots compared to 4-day-old roots, however, an increase was not visible in 6-day-old roots compared to 5-day-old roots (Figures 1d and 3a). We found that the rate of CEI divisions within our model was influenced by the QC division. For example, the change in WOX5 expression upon QC division impacts SHR expression and thus indirectly the SHR/SCR complex formation. The SHR/SCR complex, in turn, directly regulates *CYCD6;1* expression, which triggers CEI divisions. As such, CEI divisions are temporally correlated with the QC divisions. To test the involvement of protein movement in the interdependence of QC and CEI divisions, we quantified the CEI divisions in a *wox5*xpWOX5:WOX5-3xGFP line where WOX5 movement is inhibited (Berckmans et al., 2020). The number of divided CEIs was reduced in the *wox5*xpWOX5:WOX5-3xGFP line, potentially the result of WOX5 repressing activities on SHR in the vascular

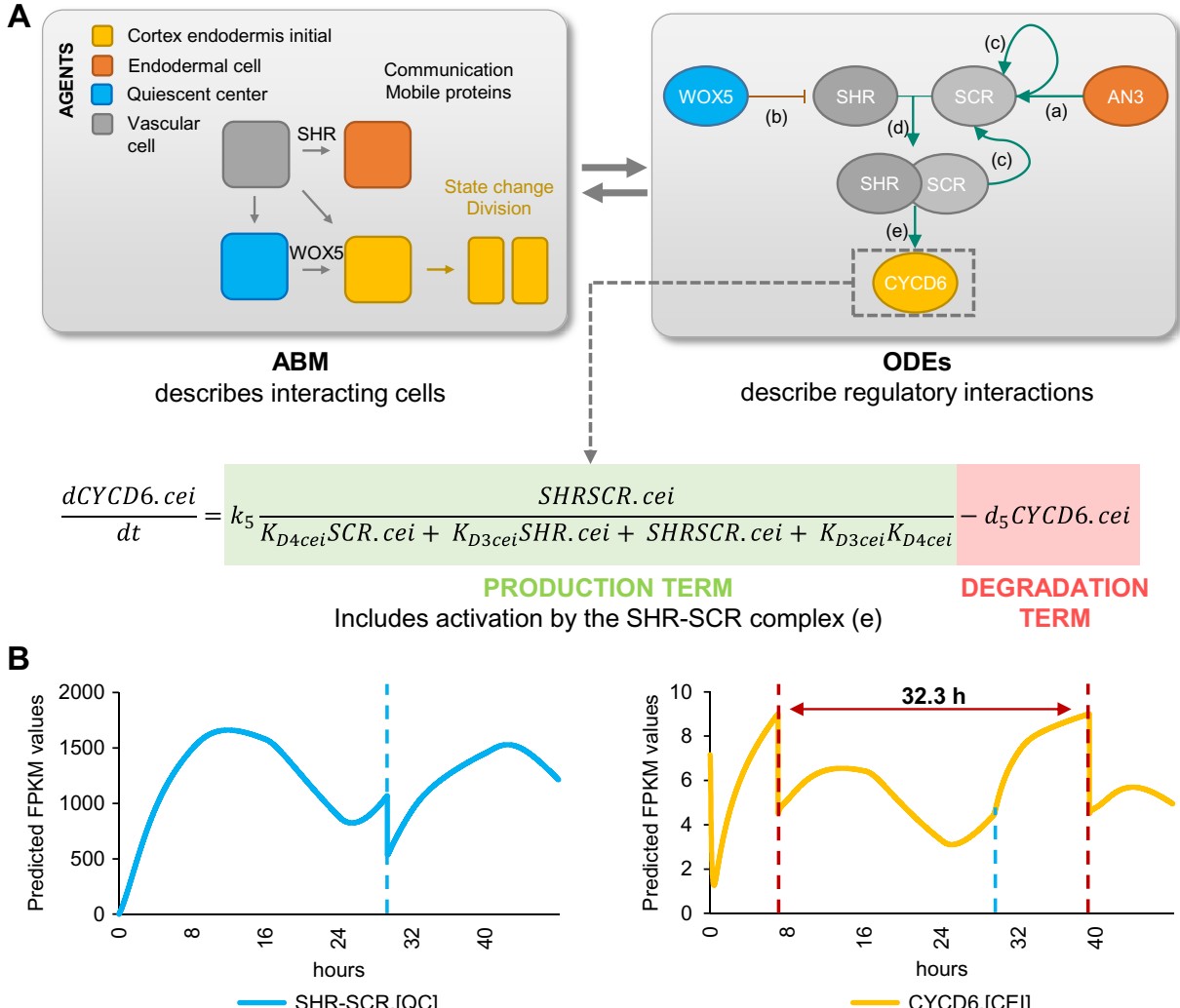

**Fig. 4.** Computational hybrid modeling of quiescent center (QC) and cortex endodermis initial (CEI) division behaviour. (a) A hybrid model combines agent-based model (ABM) rules with ordinary differential equations (ODEs). Left panel: four cell types are considered as the agents in the model interacting with each other through mobile proteins and changing states through cell division. Right panel: known regulatory interactions between key genes involved in regulating CEI division [a: (Ercoli et al., 2018), b: (Clark, Fisher, et al., 2020), c: (Helariutta et al., 2000), d: (Long et al., 2017), e: (Sozzani et al., 2010)]. (b) Model simulation of the expression of SHORTROOT (SHR)/SCARECROW (SCR) complex and D-type Cyclin CYCLIND6;1 (CYCD6;1) in the QC and CEI, respectively. Red dotted lines indicate CEI divisions and the blue dotted line indicates the time point of the QC division.

initials (Supplemental Figure 5e,f) and, accordingly, reduced levels of SHR decreases *CYCD6;1* activation in the CEIs (Koizumi et al., 2012). The distinct phenotype of *wox5*xpWOX5:WOX5-3xGFP line compared to the *wox5* mutant, which showed an increased number of divided CEIs, and the complemented *wox5*xpWOX5:WOX5-xGFP suggested that WOX5 movement is important for proper CEI divisions. Taken together, our results suggest a QC division at 5 days 5 hours resulting from high SHR/SCR and low WOX5 concentrations, CEI divisions at 4 days 8 hours and 5 days 16 hours resulting from high CYCD6;1 concentrations, and an interdependence between CEI divisions and QC divisions.

## 2.5. The hybrid model partially captures systems behaviour in response to molecular perturbations

The regulatory network underlying the hybrid model can recapitulate the QC and CEI divisions in WT conditions. However, to further validate the model, we simulated the loss-of-function of *wox5* and *an3* and evaluated the expression patterns as well as CEI

division dynamics. Based on transcriptome data of *wox5* and *an3*, we calculated 99.53% and 88.12% reduction of *WOX5* and *AN3* expression in their respective loss-of-function lines (Supplemental Figure 7). As such, the initial expression levels of WOX5 and AN3 were set to 0.47% and 11.88% in the mutant simulation as compared to the values in a WT situation, respectively.

Model simulations of *wox5* loss-of-function predicted an additional CEI division between 4 and 5 days compared to WT, which coincided with an increase in divided CEI cells at 4 days in *wox5* (Supplemental Figures 8a and Figure 1d). The additional division is most likely the result of the removal of WOX5 repression on *SHR* in the vascular initials leading to an accelerated accumulation of SHR/SCR complex in the CEI. An overall increase in SHR/SCR in the CEI was not predicted by the model (Supplemental Figure 9b), and accordingly, CEI-specific transcriptomics and protein quantifications in the CEI of the *wox5* mutant did not show an increased *SHR* expression (Supplemental Table 2, Supplemental Figure 9a). The simulations of the *an3* loss-of-function predicted the depletion of SCR in the QC, CEI and endodermal cell compared to WT

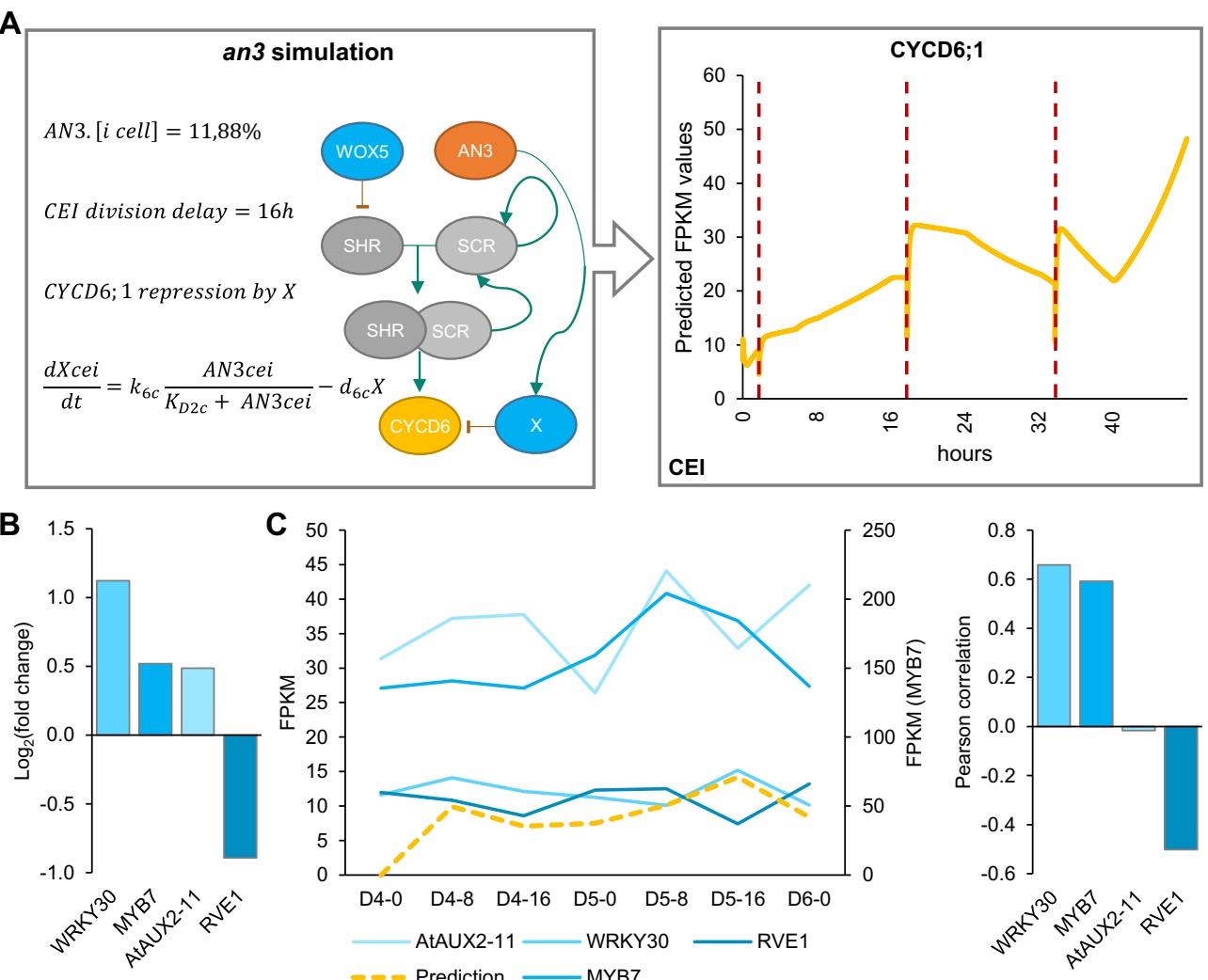

**Fig. 5.** Mathematical modeling of CEI behaviour in the angustifolia 3 (*an3*) mutant background. (a) Left panel indicates the modifications made to the model. Right panel shows the *CYCD6;1* expression during the *an3* simulation in the cortex endodermis initial (CEI) agent. Red dotted lines indicate a division. (b) The expression values of transcriptional repressors within the *an3* transcriptome dataset identified through overlap with the TOPLESS interactome. (c) The expression of the four identified transcriptional repressors in the *an3* mutant within the stem cell time course (left) and Pearson correlation with the model predicted FPKM (fragments per kilobase per million mapped reads) values (right).

(Supplemental Figure 8b). This decrease in *SCR* expression has been shown within the QC (Ercoli et al., 2018). However, the CEI and endodermis still showed high levels of SCR when a repressor version of AN3 is expressed in the SCR reporter line (Ercoli et al., 2018), which is in contrast to the model predictions. As such, the regulation of *CYCD6;1* by AN3 in the CEI may not be established via SCR but another unknown mechanism. We hypothesized that AN3 is regulating an additional factor that represses *CYCD6;1*. For this, we added an unknown factor X that is activated by AN3 and represses *CYCD6;1*, removed the AN3 activation of *SCR*, updated the ODEs within the CEI agent accordingly, and re-estimated four former and two new parameters (see Section 4) (Supplemental Tables 7 and 8). During model optimization, an additional rule that ensured a fixed minimum time between two CEI divisions was implemented to overcome overproliferation in the model (see Section 4). By adding competition between a repressor, transcriptionally activated by AN3, and the SHR/SCR direct regulation of *CYCD6;1*, the model was able to accurately capture the CEI divisions in a wild-type situation as well as in an *an3* mutant background (Figure 5a). Notably, by adding the repressor to the

model, the CEI division time interval shortened to 23.3 hours (Supplemental Figure 10). To identify potential candidates as a repressor downstream of AN3, we performed genome-wide expression analysis on *an3* meristematic root tissue (Supplemental Table 9). In total 1013 genes were differentially expressed (FDR < .05) including 67 TFs of which four TFs shown to interact with TOPLESS, a known transcriptional co-repressor (Causier et al., 2012) (Figure 5b). Of these four transcriptional repressors, WRKY30 and MYB7 showed the highest expression correlation with the model prediction (Figure 5c). WRKY30 and MYB7 were also identified as a downstream target of AN3 in a tandem chromatin affinity purification (TChAP) experiment (Vercruyssen et al., 2014). AtAUX2-11 and RVE1 showed no correlation and anti-correlation with the model predictions, respectively. As such, we propose WRKY30 or MYB7 as the putative downstream target of AN3 and repressor of *CYCD6;1* in the model. Our hybrid model suggests that the regulation of CEI divisions by AN3 does not occur through its regulation of *SCR*. Model predictions propose an unknown repressor activated by AN3 that is able to control *CYCD6;1* expression. Overall, we modelled systemic behaviour and

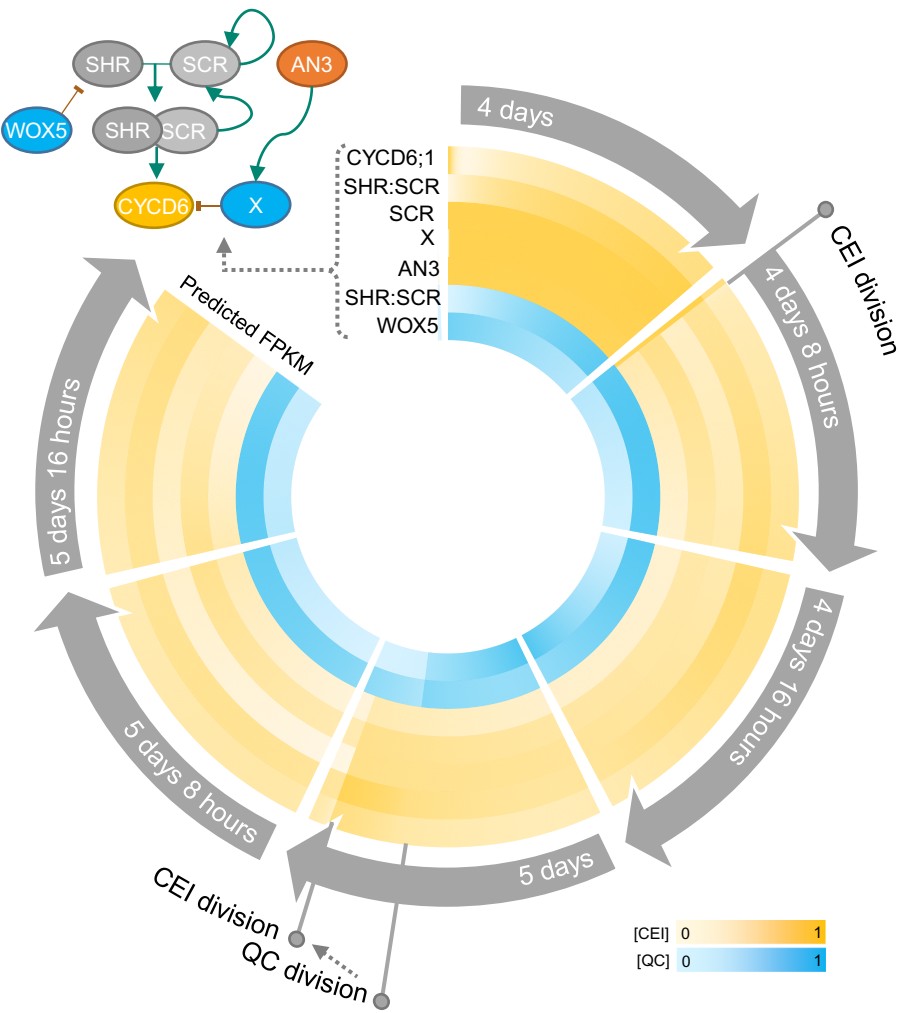

**Fig. 6.** Overview of stem cell division dynamics. Circular heatmap of the predicted scaled FPKM values of *WOX5*, *SHR/SCR*, *AN3*, CYCD6;1-repressor *X*, *SCR*, and *CYCD6;1* over time from 4-day-old roots up to 6-day-old roots. The yellow and blue shades are predicted FPKM values in the cortex endodermis initial (CEI) and the quiescent center (QC), respectively. CEI and QC divisions are marked upon the heatmap. On the left of the circular heatmap, a network with the known and predicted regulatory interactions between these key proteins is drawn. Green and red arrows represent activation and repression, respectively.

predicted SCR, SHR, WOX5, AN3 and CYCD6;1 cell-type-specific protein concentrations as well as QC and CEI division dynamics.

## 3. Discussion

Plants are multi-scale systems, in which cellular processes, such as the divisions of cells, occur at a different timing than molecular processes, such as Figure 6 protein movement. To understand these multi-scale systems and connect molecular dynamics with phenotypic changes, models that take into account multiple scales are becoming increasingly important. We have proposed an ODE and agent-based hybrid model that allows for the exchange of information across biological scales, from a molecular scale (i.e., regulatory interactions at single cell level) to a cellular scale (i.e., division of stem cells). As such, protein abundances have a direct influence on cell divisions and vice versa. Additionally, the cell divisions within the model could be triggered by the expression dynamics of regulatory networks within each cell.

In the Arabidopsis root SCN, the different stem cell types and the QC are positioned in a highly regular and well-characterized organization. The asymmetric divisions of these organized stem

cells form all cell- and tissue-types of the Arabidopsis root and are controlled by dynamic, yet robust, regulatory signalling mechanisms. Several TFs have been identified in a cell-type specific context to regulate stem cell divisions. For example, SHR and SCR are known to activate *CYCD6;1* in the CEI (Sozzani et al., 2010) and, in this study, we propose a non-cell autonomous function for WOX5 in the regulation of CEI divisions. We cannot exclude that auxin is also involved in regulating CEI divisions, as the presence of an auxin maximum in the root correlates with and positively influences *CYCD6;1* expression in the CEI (Cruz-Ramírez et al., 2012). Accordingly, it has been shown that treating plants with auxin results in an extended *CYCD6;1* expression and the presence of additional periclinal divisions (Cruz-Ramírez et al., 2012). However, the function of key proteins, such as WOX5 and SHR, on a system-level scale is unknown and key questions remain: How do key regulatory proteins coordinately regulate stem cell divisions? What set of rules and parameters govern these complex systems? In this study, we have used a multi-scale hybrid model to advance research that aims to connect molecular dynamics with phenotypic changes. The connection between regulatory inputs and cellular behaviour, such as cell division, is highly complex and

requires computational models to generate and test hypotheses about the rules governing these cellular behaviours. The hybrid model allowed us to describe complex systemic behaviour by combining: (a) discrete agent-based modeling aspects to incorporate cell-specificity and allow for cell divisions through simple rules and (b) continuous ODE models to describe the expression dynamics of the included proteins (Figure 6). Including interactions between agents/cells is critical to fully address system-level problems and replicate observable behaviours. Questions about how mobile proteins affect phenotypic changes can be addressed by instructing agents/cells to communicate effectively in a model. To note, this model is not attempting to simulate and predict the division plane or direction. The ODE and agent-based hybrid model includes short range signals allowing for cell-to-cell communication. The mobile proteins, WOX5 and SHR, non-cell-autonomously regulate the expression of downstream proteins in specific cell types and allow for the communication between these cell types. WOX5 proteins can move to the neighbouring vascular initials and CEI cells and SHR proteins move to the QC, CEI and endodermal cells. Scanning FCS was used to quantify the diffusion coefficient of WOX5 and SHR to include into the model (Supplemental Table 7) (Clark et al., 2016; Clark, Van den Broeck et al., 2020). As such, the model predicted an additional CEI division in *wox5* mutant as a result of the non-cell-autonomous regulation of *SHR* by WOX5 in the vascular initials and the movement of SHR to the CEI. Importantly, the inclusion of cell-to-cell communication into the model was crucial to accurately model stem cell division dynamics and contributed towards a better understanding of the rules underlying cellular behaviour (Figure 6).

Overall, our computational models and approach were aimed at making predictions about the rules of stem cell divisions that lead to testable hypotheses and assist in making future decisions. Accordingly, since the model suggested that the CEI-specific role of AN3 was not established through the regulatory interaction with SCR, we implemented a transcriptional repressor regulated by AN3, a non-intuitive aspect, to simulate the additional CEI divisions as found in an *an3* background (Figure 6). Four candidate transcriptional repressors (Causier et al., 2012) downstream of AN3 and upstream of *CYCD6;1* were proposed based on transcriptome analysis, of which *WRKY30* and *MYB7* showed the highest correlation with model predictions and were identified as a downstream target of AN3 in a TChAP experiment (Vercruyssen et al., 2014). Even though, since this is outside the scope of the study, the roles of these four TFs in regulating stem cell division within the SCN remain elusive, our integrative multi-scale model allowed us to both (a) predict cellular behaviour in normal conditions and (b) capture CEI division dynamics in response to perturbations. Thus, by combining continuous models to describe cell-specific regulatory networks and agent-based rules, systemic behaviour was modelled and led to a deeper understanding of the regulatory rules governing cell division.

# 4. Materials and methods

## 4.1. Plant material and growth conditions

The *wox5* and *an3* loss-of-function lines, pAN3:AN3-GFP, 35S-AN3-GFP, pWOX5:WOX5-GFP, *wox5* x pWOX5:WOX5-3xGFP, pCYCD6;1:GUS-GFP and *wox5* x pCYCD6;1:GUS-GFP are previously described in various studies (Berckmans et al., 2020; Clark, Fisher, et al., 2020; Ercoli et al., 2018; Sozzani et al., 2010; Vercruyssen et al., 2014). *an3* x pCYCD6;1:GUS-GFP was gen-

erated by crossing *an3* with pCYCD6;1:GUS-GFP. Homozygous plants were selected by PCR using the SALK LB primer and the AN3-specific oligos 5'-ATTACGACACAACTTGGAGCC-3' and 5'-TTTGTGGTCCGAAACAACATC-3'. All lines were upscaled with their corresponding wild type.

For imaging and root growth assays, seeds were dry sterilized using fumes produced by a solution of 100% bleach and 1M hydrochloric acid. The seeds were plated on square Petri dishes with solid (10 g/L agar, Difco$^{TM}$) 1X MS (Murashige and Skoog) medium supplemented with 1% sucrose and stratified for 2 days at 4°C. The plates were grown vertically at 22°C in long-day conditions (16-hrs light/8-hrs dark) for 4, 5, 6, or 7 days as indicated in the figures. At least three biological replicates of 10–20 plants were performed for the root growth assays and confocal images. The different lines were always grown together on one plate with the appropriate control line. For RNAseq experiments, seeds were wet sterilized using 50% bleach, 100% ethanol and water. Seeds were imbibed and stratified for 2 days at 4°C. Next, the seeds were plated with high density on Nitex mesh squares on top of solid 1X MS medium with 1% sucrose. Seeds were plated and grown vertically at 22°C in long-day conditions.

## 4.2. Root growth assays

At 3, 4, 5, 6 and 7 days, the primary root length was marked. At 7 days, a picture of the marked square plates was taken and the root length was measured using the software program ImageJ version 1.45 (National Institutes of Health; http://rsb.info.nih.gov/ij/). For the statistical analysis of the root growth assays, Student's *t*-tests were performed on the average of each biological replicate.

## 4.3. Confocal imaging, pair correlation function analysis and number and brightness

Confocal microscopy was conducted using a Zeiss LSM 710 or 880 on 4-, 5-, or 6-day-old root tips. The 488 nm and 570 nm lasers were used for green and red channel acquisition, respectively. Propidium iodide (10 μM, Calbiochem) was used to stain cell walls and mPS-PI (modified pseudo-Schiff-PI) staining was used to visualize starch granules. For the N&B acquisition, 12-bit raster scans of a 256 × 256 pixel region of interest were acquired with a pixel size of 100 nm and a pixel dwell time of 12.61 μs as described in Clark et al., 2016; Clark & Sozzani, 2017. For pair correlation function (pCF) acquisition, 100,000 12-bit line scans of a 32 × 1 pixel region of interest were acquired with a varying pixel size and a pixel dwell time of 8.19 μs as described in Clark et al., 2016; Clark & Sozzani, 2017. Heptane glue was used during N&B and pCF acquisition to prevent movement of the sample as described in Clark et al., 2016; Clark & Sozzani, 2017.

Analysis of confocal images for CTCF measurements was performed as described previously (Clark et al., 2019). Analysis of the raster scans acquired for N&B and the line scans for pCF was performed using the SimFCS software (https://www.lfd.uci.edu/globals/). For N&B, the 35S:GFP line was used to normalize the background region of the image (S-factor of 2.65) and determine monomer brightness (brightness of 0.26). A 128 × 128 region of interest was used on all images to measure oligomeric state specifically in the QC. For pCF, each line scan image was analysed with three different pixel distances (8, 10 and 12, or 7, 9 and 11) in both a left-to-right (movement from QC to CEI) and a right-to-left scanning direction (movement from CEI to QC). For each technical replicate of a line scan image, a qualitative Movement Index (MI)

was assigned based on the detection of movement in the carpet (arch pattern, MI = 1) or not (no arch pattern, MI = 0) (Clark et al., 2016; Clark & Sozzani, 2017). The technical replicates were then averaged for each biological replicate. The pWOX5:WOX5:GFP images were analysed separately in both directions.

### 4.4. RNAseq analysis and network inference

300–500 mg of pWOX5:erGFP, pCYCD6:GUS-GFP and wox5xp CYCD6:GUS-GFP seeds were wet sterilized and plated for each of the four biological replicates. After 5 days of growth, approximately 1 mm of the root tip was collected and protoplasted as described (Birnbaum et al., 2005). GFP positive and negative cells were collected using a MoFlo cell sorter into a vial containing a solution of beta-mercaptoethanol and RLT buffer. RNA was extracted using the Qiagen RNeasy Micro kit. Libraries were prepared using the SMART-Seq v3 Ultra Low RNA Input Kit for Sequencing and the Low Library Prep Kit v1 from Clontech. For the *an3* RNAseq experiment, ~5 mm of *an3* and WT root tips were collected for each of the three biological replicates. RNA was extracted using the Qiagen RNeasy Micro kit and libraries were prepared using the NEBNext Ultra II RNA Library Prep Kit for Illumina (New England BioLabs). All libraries were sequenced on an Illumina HiSeq 2500 with 100 bp single-end reads.

Gene expression analysis of raw RNA-seq data and subsequent GRN inference was performed using the TuxNet interface (Spurney et al., 2019). Specifically, TuxNet uses ea-utils fastq-mcf (Aronesty, 2011; 2013) for pre-processing, hisat2 (Kim et al., 2015) for genome alignment and Cufflinks (Trapnell et al., 2012) for differential expression analysis. To infer a gene regulatory network (GRN) and predict the causal relationships of genes regulating CEI identity, DEGs were identified using FDR < .05 as our selection criteria, when performing pairwise comparisons between GFP negative cells from pWOX5:erGFP and GFP positive cells from pCYCD6:GUS-GFP or *wox5* x pCYCD6:GUS-GFP. Within the TuxNet interface, RTP-STAR (Regression Tree Pipeline for Spatial, Temporal and Replicate data) was used for all network inference. The pipeline consists of three parts: spatial clustering using the *k*-means method, network inference using GENIE3 and edge sign (activation or repression) identification using the first-order Markov method. TuxNet is available at https://github.com/rspurney/TuxNet and video tutorials regarding installation, analysis and network inference are freely available at https://rspurney.github.io/TuxNet/. The network was visualized in Cytoscape® 3.8.0 (Shannon et al., 2003).

### 4.5. Node impact analysis

Each node from the network receives a weight between 1 and 2:

$$weight\,(N) = w = 1 + \frac{O}{O_{max}}$$

Nodes with a high outdegree (O) are considered to be more impactful within the network and will thus receive a high weight. The impact of a node within the network topology is calculated based on the weighted first neighbours:

$$R = ASPL \times \sum_{1\,to\,O}^{i} w_i + A \times \sum_{1\,to\,I}^{i} w_i$$

$$A = \frac{Nodes\,(outdegree > 0)}{Nodes}$$

where $R$ = Robustness, $ASPL$ = Average Shortest Path Length, $O$ = outdegree and $I$ = indegree. A scale-free network will have a low $A$, while a scale-rich network will have a high $A$, allowing for the indegree to contribute more to the impact of a node. Because the first neighbours are weighted in regards to their outdegree, genes with a lower outdegree can still have a large impact if its neighbours have a high outdegree and the gene is thus centrally located. Genes with a large number of cascading targets that are two or more nodes away will have a higher ASPL and thus a higher scaled outdegree weight, accurately reflecting the hierarchical importance of the source gene itself and its first neighbours targets.

### 4.6. Shiny app: node analyzer

To calculate necessary network statistics such as outdegree and indegree in Cytoscape® 3.8.0 (Shannon et al., 2003), select Tools -> Analyze Network, check the Analyze as Directed Graph if applicable, and then press OK to perform the analysis. To export node and edge files from Cytoscape, select File -> Export -> Table to File, and then choose default edge or default node in the 'Select a table to export' dropdown. Press OK to export each file. Import the node and edge table files into the corresponding prompts (Figure 2c) and press the Run Analysis button to calculate impact scores. Results can be downloaded as a table using the Download Results button. In addition to the impact scores, the application renders three plots for visualization: one plot with the impact score for each gene and two histograms with the indegree and outdegree.

The Node Analyzer user interface can be accessed online at https://rspurney.shinyapps.io/nodeanalyzer/ or ran through R with scripts freely available at https://github.com/rspurney/NodeAnalyzer. Example datasets are also available via the GitHub link.

### 4.7. Ordinary differential equations, parameter estimation, and sensitivity analysis

ODEs were developed to model the dynamics of CYCD6;1, its upstream regulators SHR and SCR, WOX5 and AN3 in three different cell types: endodermal cell, CEI, and QC. The regulatory interactions between these five proteins were modelled using Hill equation dynamics and SHR/SCR complex formation is modelled using mass-action kinetics. SHR and WOX5 diffusion are modelled using a linear term for gradient-independent diffusion. All proteins are assumed to have a linear degradation term. We modelled transcriptional regulation and protein expression in the same equation.

(1) **SHR**; for the upstream regulation of *SHR* in the vasculature, the repression by WOX5 was included (top equation) (Clark, Fisher, et al., 2020).

$$\frac{dSHR.[vasc]}{dt} = k_4 \frac{K_{D1vasc}}{K_{D1vasc} + WOX5.[vasc]} - d_4 SHR.[vasc]$$

(2) **SCR**; for the upstream regulation of *SCR* expression, we included the autoactivation by SCR itself (Cruz-Ramírez et al., 2012; Heidstra et al., 2004), the activation by the

SHR/SCR complex (SSC) (Heidstra et al., 2004) and the activation by AN3 (Ercoli et al., 2018). Each one of these regulations was assumed to be sufficient to induce *SCR* expression.

$$\frac{dSCR.\left[i\,cell\right]}{dt} =$$

$$k_{3i}\left(\frac{K_{D4i}SCR.\left[i\,cell\right]+SSC.\left[i\,cell\right]}{K_{D3i}K_{D4i}+K_{D4i}SCR.\left[i\,cell\right]+K_{D3i}SHR.\left[i\,cell\right]+SSC.\left[i\,cell\right]}\right.$$
$$\left.+\frac{AN3.\left[i\,cell\right]}{K_{D2i}+AN3.\left[i\,cell\right]}\right)-d_{3i}SCR.\left[i\,cell\right]$$

(3) **WOX5**; the production of WOX5 was assumed to be time-dependent as this produces the best model fit to the experimental data (top equation) (Clark, Fisher, et al., 2020).

$$\frac{dWOX5.\left[QC\right]}{dt} = k_{1qc}WOX5.\left[QC\right]$$

(4) **AN3**; the production of AN3 was assumed to be time-dependent as this produces the best model fit to the experimental data.

$$\frac{dAN3.\left[i\,cell\right]}{dt} = k_{2i}AN3.\left[i\,cell\right]$$

(5) **CYCD6;1**; for the upstream regulation of *CYCD6;1* expression, we included the activation by the SHR/SCR complex (SSC) (Sozzani et al., 2010).

$$\frac{dCYCD6.\left[CEI\right]}{dt} =$$

$$k_{5}\frac{SSC.\left[CEI\right]}{K_{D4cei}SCR.\left[CEI\right]+K_{D3cei}SHR.\left[CEI\right]+SSC.\left[CEI\right]+K_{D3cei}K_{D4cei}}$$
$$-d_{5}CYCD6.\left[CEI\right]$$

It was shown that the different oligomeric forms and stoichiometries of SHR, SCR and the SHR/SCR complex show a similar expression pattern (Clark, Fisher, et al., 2020). As such, the SHR and SCR oligomeric forms were modelled as one variable.

The interaction between the different agents/cell types is modelled using mass-action kinetics. The state change following division is modelled using simple agent-based rules. To simulate division of an agent, the capacity of the cell doubles, subsequently halving all proteins present.

(6) The cell types interact with each other through the movement of the regulatory proteins SHR and WOX5. The amount of SHR in the other cell types was determined by the movement of SHR (top equation). The amount of WOX5 in the vasculature was determined by the movement of WOX5 from the QC (bottom equation) (Figure 1).

$$\frac{dSHR.\left[i\,cell\right]}{dt} = a_{i}SHR.\left[vasc\right]-d_{12i}SHR.\left[i\,cell\right]$$

$$\frac{dWOX5.\left[vasc\right]}{dt} = a_{vasc}WOX5.\left[QC\right]-d_{1vasc}WOX5.\left[vasc\right]$$

(7) It was shown that the division of the QC cell correlates with the expression of WOX5 and the SHR/SCR complex (SSC)

(Clark, Fisher, et al., 2020).

$$if\ WOX5.\left[QC\right]\le 100\,\&\,SSC.\left[QC\right]\le 1100:\frac{Gene_{0\ to\ j}.\left[QC\right]}{2}$$

(8) We assumed that the division of the CEI cells is dependent on the expression of CYCD6;1 (Sozzani et al., 2010).

$$if\ CYCD6.\left[CEI\right]\ge 9:\frac{Gene_{0\ to\ j}.\left[CEI\right]}{2}$$

For the sensitivity analysis, the total Sobol effect index was calculated for each parameter value (Saltelli et al., 2010; Sobol', 2001). Parameter values were randomly sampled using Monte Carlo sampling to obtain 150 different values for each parameter. This analysis was repeated for 10 technical replicates. As such, for each parameter 170 (10 replicates × 17 ODEs) total Sobol effect indices were obtained. For each ODE and replicate, the sensitivities were rescaled between 0 and 1 and then averaged across the 17 ODEs. The obtained averaged sensitivities for each replicate were again averaged to retrieve the total Sobol effect index per parameter (Supplemental Table 4). The sensitive parameters were chosen as the parameters that had significantly higher Sobol indices than the lowest scoring parameter (K_D2_qc) using a Student's *t*-test ($p < .01$).

To estimate the sensitive parameters, the model was fitted onto extrapolated cell-type specific time course expression data (Supplemental Table 5). To generate this cell-types specific time course expression data, FPKM (fragments per kilobase per million mapped reads) values in the QC, CEI and vascular initials at 5 days were obtained from Clark et al., and the endodermis-specific FPKM values at 5 days were obtained from Li et al (Clark et al., 2019; Li et al., 2016). Using the fold changes of a time course dataset from the root stem cell niche every 8 hours from 4 to 6 days (Clark et al., 2019) and the FPKM values at 5 days for the specific cell types, we were able to extrapolate cell-type specific time course expression values (Supplemental Table 5). Simulated annealing and Latin hypercube sampling as described in (Clark, Fisher, et al., 2020) produced 40 sets of parameter estimates (Supplemental Table 6). The average of these parameter estimates was used for the model simulations. The remaining sensitive parameters were set to a constant value from the corresponding estimated parameter in Clark, Fisher, et al., 2020. The value of non-sensitive parameters was selected based on similar values of the model described in Clark, Fisher, et al., 2020. The production terms for WOX5 (k1_qc) and AN3 (k2_qc, k2_cei and k2_endo) were set to a constant value at each time point to minimize the error between the model and the time course expression data. The diffusion coefficients of SHR (a_qc and a_cei) and WOX5 (b_qc) were experimentally determined from RICS experiments (Supplemental Table 3) (Clark, Fisher, et al., 2020).

The following changes were made in the regulatory network underlying the CEI divisions to reflect the *an3* loss-of-function in the hybrid model:

(1) **Factor X**; for the upstream regulation of the unknown repressor X in the CEI agent, the activation by AN3 was included.

$$\frac{dX.\left[CEI\right]}{dt} = k_{6cei}\frac{AN3.\left[CEI\right]}{K_{D2cei}+AN3.\left[CEI\right]}-d_{6cei}X.\left[CEI\right]$$

(2) **CYCD6;1**; for the upstream regulation of *CYCD6;1* expression, we added the repression of factor X in addition to the activation by the SHR/SCR complex (SSC) (Sozzani

et al., 2010).

$$\frac{dCYCD6.[CEI]}{dt} =$$

$$k_5\left(\frac{SSC.[CEI]}{K_{D4cei}SCR.[CEI] + K_{D3cei}SHR.[CEI] + SSC.[CEI] + K_{D3cei}K_{D4cei}} \right.$$

$$\left. + \frac{K_{D6cei}}{K_{D6cei} + X.[CEI]}\right) - d_5 CYCD6.[CEI]$$

(3) **SCR**; for the upstream regulation of *SCR* expression in the CEI and endodermal agent, we included the autoactivation by SCR itself (Cruz-Ramírez et al., 2012; Heidstra et al., 2004), the activation by the SCR-SHR complex (Heidstra et al., 2004) and removed the activation by AN3 (Ercoli et al., 2018).

$$\frac{dSCR.[i\,cell]}{dt} =$$

$$k_{3i}\frac{K_{D4i}SCR.[i\,cell] + SSC.[i\,cell]}{K_{D3i}K_{D4i} + K_{D4i}SCR.[i\,cell] + K_{D3i}SHR.[i\,cell] + SSC.[i\,cell]}$$

$$- d_{3i}SCR.[i\,cell]$$

(4) To avoid uncontrollable division within the CEI, the CEI agent was subjected to an additional rule that ensured a minimum time of 16h between successive divisions ($\Delta t$).

$$if\ CYCD6.[CEI] \geq 9 \& \Delta t > 16 : \frac{Gene_{0\,to\,j}.[CEI]}{2}$$

Four existing parameters (k3_endo, d3_endo, k3_cei and k5_cei) and two new parameters (k6_cei and d6_cei) were re-estimated in the same manner as described above and produced 20 sets of parameter estimates (Supplemental Table 8). For the remaining parameters, the same value as the initial hybrid model was used.

All parameters for the initial and adjusted model are listed in supplemental Table 7. To simulate the hybrid models, the initial values were set as the 4D FPKM values from the extrapolated time course data. For factor X, the SHR/SCR complex, and very lowly expressed genes (e.g., WOX5 in the vascular initials) the initial value was zero. To simulate *wox5* loss-of-function, the initial value of WOX5 was set to 0.47% (Supplemental Figure 7). To simulate *an3* loss-of-function, the initial value of AN3 in all three agents was set to 11.88% (Supplemental Figure 7). ODE45 was used as the ODE solver within SimBiology.

### Acknowledgements

We thank Dr. Kensuke Kawade for *an3* seeds, Dr. Javier F. Palatnik for the pAN3:AN3-GFP seeds, Dr. Dirk Inzé for the p35S:AN3-GFP seeds, Dr. Rüdiger Simon for the *wox5* x pWOX5:WOX5-3xGFP and Dr. Thomas Laux for pWOX5:WOX5-GFP seeds.

**Financial support.** This work was supported by the National Science Foundation (NSF) (CAREER MCB-1453130) to RS; Foundation for Food and Agriculture Research (FFAR) to RS; and NSF/Biotechnology and Biological Sciences Research Council (BBSRC) (MCB-1517058) to TAL and RS.

**Conflict of interest.** The authors declare no conflict of interest.

**Authorship contributions.** L.V.d.B., A.P.F. and R.S. conceived and designed the study. L.V.d.B. and R.J.S. conducted the computational modeling. N.M.C. advised on the modeling. R.J.S designed the Shiny App. A.P.F., L.V.d.B., T.T.N, I.M and M.G gathered experimental data. L.V.d.B. and A.P.F. analysed experimental data. L.V.d.B. performed statistical analysis. L.V.d.B., A.P.F, R.J.S. and R.S. wrote the manuscript and all authors contributed to correcting the manuscript.

**Data availability statement.** All sequencing data are available on GEO at:
- https://www.ncbi.nlm.nih.gov/geo/query/acc.cgi?acc=GSE155462
- https://www.ncbi.nlm.nih.gov/geo/query/acc.cgi?acc=GSE155463

MATLAB code used for the hybrid model is available at https://github.com/LisaVdB/Hybrid_model_CEI_division. R-code used to develop the Shiny application is available at https://github.com/rspurney/NodeAnalyzer.

**Supplementary Material.** To view supplementary material for this article, please visit http://dx.doi.org/10.1017/qpb.2021.1.

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
