## [Reviewer Report]

Dear Editor,

We wish to submit an original research manuscript entitled “Exchange of molecular and cellular information: a hybrid model that integrates stem cell divisions and key regulatory interactions” for consideration by Quantitative Plant Biology journal. 

Stem cells undergo asymmetric divisions to give rise to all cell types in an organ. The transcriptional interactions regulating these stem cell divisions are determinant for proper organ development and growth. However, these regulatory mechanisms are complex and dynamic and require quantitative analyses over time and space coupled with computational models. In this manuscript, we combined cell-type-specific gene expression data and experimental data, such as scanning fluorescence correlation spectroscopy, with mathematical models to better understand how key regulatory proteins coordinately regulate stem cell divisions. We constructed a multi-scale model that, through the use of simple rules, can simulate quantitative protein concentration dynamics and system-level behavior. Specifically, we coupled agent-based modeling aspects and continuous models of ordinary differential equations to allow for the exchange of information across biological scales, from a molecular scale (i.e. regulatory interactions at single cell level) to a cellular scale (i.e. division of stem cells). As such, protein concentrations have a direct influence on cell divisions and vice versa. We included four stem cell types of the Arabidopsis thaliana root stem cell niche in the hybrid model: quiescent center (QC), vascular initial, cortex epidermis initial (CEI), and endodermal cell. Additionally, we included mobile proteins, WOX5 and SHR, that non-cell-autonomously regulate the concentration of downstream proteins in specific cell types. The communication between the cell types and the dynamic expression patterns of key proteins, such as WOX5, AN3, SCR, SHR, and CYCD6;1, allowed the hybrid model to predict experimentally validated stem cell divisions.

Given the increasing amount of data generated in the science community, methodologies to analyze and interpret this data are becoming more important. In this manuscript, we have developed a shiny R application that easily allows for downstream gene regulatory network analyses, assisting biologists in interpreting complex networks. More importantly, within this manuscript a multi-scale model was developed that uses simple rules and continuous models to simulate quantitative protein dynamics and system-level behavior, such as cell divisions. As plants are dynamic and multiscale systems, models that take into account multiple scales will become increasingly important. We are confident that gene regulatory networks and multi-scale models could aid in a better understanding of quantitative plant research for many biologists. 

As this manuscript combines experimental data, such as scanning fluorescence correlation spectroscopy, transcriptomics, and phenotypical analyses, with computational tools, like network inference, node importance analysis, and mathematical modeling, it would be an excellent fit for the Quantitative Plant Biology journal. This publication is original and has not been published elsewhere, nor is it currently under consideration for publication elsewhere.

We would appreciate your willingness to consider this manuscript for publication in Quantitative Plant Biology. Please feel free to contact us if you have any questions or comments. 

Sincerely,

Lisa Van den Broeck and Ross Sozzani

---

## [Reviewer Report]

*Comments to Author*: In the manuscript entitled: ¨Exchange of molecular and cellular information: a hybrid model that integrates stem cell divisions and key regulatory interactions¨ by Van den Broeck etal, authors describe the capacity of WOX5 protein movement from QC cells to CEI cells. They also correlate the altered expression of *CYCD6;1* promoter with extra ACDs in the *wox5* mutant background and, based on such observations they propose a non-cell autonomous function of WOX5 in modulating ACDs of CEI cells.

Based on such observations authors performed RNA-seq analysis of pCYCD6;1-GFP GFP-positive sorted cells from Wt and *wox5* backgrounds, as wells of pWOX5:GFP-positive sorted cells in Wt.

Several *in silico* approaches led them to identify a novel player in the SHR-SCR-CYCD6 pathway that controls ACDs in the CEI lineage. Authors integrate both WOX5 and ANGUSTIFOLIA (AN3) / GRF-INTERACTING FACTOR 1 118 (GIF1) in a more complex pathway.

Authors finally develop a hybrid model that combines agent-based modeling aspects with Operational differential equations. Their mode predicted that additional CEI ACDs in wox5 mutant are the result of on-cell-autonomous regulation of SHR by WOX5 in the vascular initials, as well as the role of AN3 acting upstream *CYCD6;1* transcription.

This study contributes with novel players in the molecular network that modulates cell reprogramming events in the ground tissue stem cell and its derived daughters. It is of great value that authors combine developmental, transcriptional and theoretical techniques to explain the complex dynamics in the modulation of ground tissue formative divisions.

However, this reviewer has several comments and suggestions, which should be solved prior full acceptance in QPB Journal.

Comments and suggestions:

**Page 5*:

84 -Expanded expression of CYCD6;1

85- in the endodermal cells resulted in additional periclinal divisions, indicating that these cells, further

86-referred to as CEI-like cells, may have gained stem cell-like characteristics (Fig 1C).

Although in Figure 1C there is an altered expression of pCYCD6;1-GFP marker, zoom squares showing extra divisions, these are occurring in the transit amplifying zone, far away from the region of action of WOX5. How can authors explain that the absence of WOX5 is altering the expression and action of CYCD6 on the ground tissue in such regions?

Moreover, how can authors discard that such pCYCD6-GFP expression pattern and divisions are not related to middle cortex formation?

Also, should acknowledge that the altered expression of CYCD6;1 correlates with extra ACDs has been previously shown in Sozzani etal 2010, Cruz-Ramirez etal 2012.

Regarding Figure 1, can authors include in this figure images of the movement of WOX5 from QC to CEI at te protein level? Is it localized in the nucleus or it is cytoplasmic?

**Page 8 Figure 2*

Panel C is not that relevant to be occupying suh a space in the figure, could it be replaced in a supplementary file?

Authors could take advantage of the new extra space to create a bigger, perhaps less gray, model of the interactions shown in panel B.

**Page 9 Figure 3*

Figure legend for this figure does not explain or cite panel C.

**On modelling and discussion sections*

**Figure 4*

Can authors elaborate prior the design of the modelling or at discussion section why previous results on WOX5 negative action overCYCD3;3 were not mentioned nor considered?

Although this regulation occurs in the QC, Forzani etal (2014) showed that in the *wox5-1* mutant, QC46 marker expands to the CEI and this expanded expression is reverted in the triple *wox5-1;cycd1,1; Cycd3;3* mutant.

A similar observation, authors should elaborate on the action of CYCD6 action on RBR and, indirectly over SCR-SHR transcriptional potential. This is quite important since there is key missing player between in the circular network in the CEI, formed by CYCD6;1-RBR-SCR/SHR-CYCD6, as modelled in Cruz-Ramírez etal 2012.

This is significant since SCR mutants that cannot bind RBR display a similar phenotype with extra ACDs in both CEI and QC lineages, which also correlate with the observed alteration in *wox5-1* mutant and CYCD3;3 (Cruz-Ramírez, etal 2013 and Forzani etal, 2014).

One would expect that if this players are not integrated in the hybrid modelling effort, at least the reason of its exclusion should be argued in the discussion section.

Finally, I strongly recommend to include a new figure with a graphic model, at the end of the article, summarizing the findings of the model and the in planta observations from previous studies and highlighting the novel players and interactions found in this study.

---

## [Reviewer Report]

*Comments to Author*: The authors described cell division dynamics in root stem cell niche in an experimental and modelling approach. Genetic analysis showed that WOX5 and AN3 repressed cell divisions in CEI via downregulation of CYCD6;1. Based on the experimental data, they integrated agent based modeling and ordinary differential equations to capture cell division behavior in QC and CEI. The model predicted the expression levels of key regulators and the correlation between QC and CEI cell divisions. Their hybrid model successfully predicted developmental dynamics in stem cell niche and provides compelling arguments.

1

The hybrid model provides the idea of interdependence of CEI and QC divisions, and the authors argued that the protein movement would be required for this interdependence. However, the importance of protein movement has not been proved experimentally. To assess the role of the protein movement, it would be better to examine the interdependence when the protein movement is inhibited (e.g. WOX5-3xGFP in wox5 mutant).

2

They calculated that the protein concentration in the daughter cells are halved after cell divisions in their model. After divisions, it is assumed that cells (agents) could be replaced by the daughter cells in some cases (for example, CEI and endodermal cells could be replaced by the QC and CEI daughter cells, respectively?). Is the replacement of the agent by the daughter cells with their properties (such as protein concentration) considered in their model? Or is this needed to be considered? Or did I misunderstood something?

3

In general, the division orientation in QC and CEI has not been mentioned in this manuscript. Does the orientation have any influence on their model?

4

L255, The authors speculated that the additive cell divisions in CEI of wox5 mutant resulted from SHR/SCR activation. However, they found that SHR expression level was not changed in CEI of wox5. Given that SHR is not transcribed in CEI, it would be good to check whether SHR protein level rather than the mRNA level is increased in CEI of wox5 or not.

5

There are some graphs showing the percentage of divided/undivided cells with error bars (e.g. Fig 1D, 3E). Is this ok to have error bars for the ratio? To my best knowledge, Chi-square test (or others) rather than t-test would be more appropriate statistical test for the ratio.

---

## [Reviewer Report]

*Comments to Author*: Dear Dr. Sozzani,

Thank you for submitting your manuscript to Quantitative Plant Biology. I read with great interest your work and the reviewers' comments, and I agree with the reviewers that the combination of experimentation and computational simulation in your study greatly advances our understanding of stem cell behavior in plants.

Nevertheless, both reviewers demand a few revisions, a small number of which may require further experimentation; I would therefore like to ask you to submit a revised manuscript that addresses those reviewers' requests. In particular, both reviewers noted that the interdependence of QC and QEI divisions relies on WOX5 protein movement, but this observation currently lacks experimental support and should be addressed — for example, by (1) showing the localization (nuclear? Cytoplasmic?) of WOX5 in QC and CEI and (2) assessing the ability of a non-mobile version of WOX5 to rescue wox5 mutant defects.

Furthermore, in addition to the revised manuscript, please upload:

(1) A point-by-point response to the reviewers' comments; please respond to all comments: if you disagree with some of them, please explain why that is so, instead of ignoring them.

(2) A version of your manuscript in which the changes made are clearly visible (e.g., a PDF of a DOC(X) file in which the changes made had been tracked with the "Track Changes" option).

I look forward to receiving your revised manuscript soon.

Sincerely yours,

Dr. Enrico Scarpella

Associate Editor

Quantitative Plant Biology